# Phenotypic and Genotypic Characteristics of Non-Hemolytic *L. monocytogenes* Isolated from Food and Processing Environments

**DOI:** 10.3390/foods12193630

**Published:** 2023-09-30

**Authors:** Barbara Szymczak

**Affiliations:** Department of Applied Microbiology and Human Nutrition Physiology, Faculty of Food Science and Fisheries, West Pomeranian University of Technology, Papieża Pawła VI 3, 71-459 Szczecin, Poland; barbara.szymczak@zut.edu.pl; Tel.: (+48)91-449-65-43

**Keywords:** atypical *Listeria monocytogenes*, antibiotic resistance, phenotyping, serotyping, sugar fermentation, virulence genes

## Abstract

Increasingly, *Listeria monocytogenes* (LM) with atypical phenotypic and genotypic characteristics are being isolated from food, causing problems with their classification and testing. From 2495 soil, food, and swab samples from the food industry, 262 LM isolates were found. A total of 30 isolates were isolated, mainly from soil and plant food, and were classified as atypical LM (aLM) because they lacked the ability to move (30/11.4%) and perform hemolysis (25/9.5%). The isolation environment affected aLM incidence, cell size, sugar fermentation capacity, antibiotic sensitivity, and the number of virulence genes. Therefore, despite several characteristics differentiating all aLMs/non-hemolytic isolates from reference LMs, the remaining phenotypic characteristics were specific to each aLM isolate (like a fingerprint). The aLM/non-hemolytic isolates, particularly those from the soil and meat industries, showed more variability in their sugar fermentation capacity and were less sensitive to antibiotics than LMs. As many as 11 (36.7%) aLM isolates had resistance to four different antibiotics or simultaneously to two antibiotics. The aLM isolates possessed 3–7 of the 12 virulence genes: *prfA* and *hly* in all aLMs, while *iap* was not present. Only five (16.7%) isolates were classified into serogroups 1/2c-3c or 4a-4c. The aLM/non-hemolytic isolates differed by many traits from *L. immobilis* and atypical *L. innocua*. The reference method of reviving and isolating LM required optimization of aLM. Statistical analyses of clustering, correlation, and PCA showed similarities and differences between LM and aLM/non-hemolytic isolates due to individual phenotypic traits and genes. Correlations were found between biochemical traits, antibiotic resistance, and virulence genes. The increase in the incidence of atypical non-hemolytic LM may pose a risk to humans, as they may not be detected by ISO methods and have greater antibiotic resistance than LM. aLM from LM can be distinguished based on lack of hemolysis, motility, growth at 4 °C, ability to ferment D-arabitol, and lack of six specific genes.

## 1. Introduction

Food-borne listeriosis is one of the most serious and severe food-borne diseases (with a high 30% rate of death) caused by the bacteria *L. monocytogenes*. In the U.S., increasing listeriosis was reported from 8 cases in 2018 to 53 cases in 2022, while 30% of the population of the EU experienced 120 cases of illness in 2020 [1]. *L. monocytogenes* is the fifth most common cause of food poisoning in the EU (2621 cases, of which 300 cases resulted in death), making *L. monocytogenes* the leading cause of food-borne fatality in Europe [2]. The most common implicated food vehicles for the strong-evidence listeriosis food-borne outbreaks were fish and fish products, ice-cream, sausages and cooked meat, fresh and ripened cheeses, fresh salads and mushrooms, and hard-boiled eggs. Currently, 29 species of *Listeria* spp. are known, of which only *L. monocytogenes* is classified as pathogenic to humans and *L. ivanovii* to animals [3,4]. Among 13 recognized serotypes of *L. monocytogenes*, only 4 are of significant public concern, with 3, 1/2a, 1/2b, and 4b, being responsible for over 95% of invasive listeriosis cases [5,6]. Most of the sporadic cases and outbreaks are associated with strains of serotype 4b, while isolates classified to serotypes 1/2a and 1/2c are more often isolated from food and environmental samples [7,8]. In addition, atypical *L. monocytogenes* serotype IV b-v1 isolated from the food production environment has been increasingly described [9].

*L. monocytogenes* is a widespread microorganism in nature, found in a wide range of animals and plants, as well as in sewage sludge, soil, and aquatic environments. Depending on the environment, aLM may acquire genes with resistance to various external agents, including antibiotics, heavy metals, or plant protection products [10].

*L. monocytogenes* is a Gram-positive, catalase-positive, and oxidase-negative bacillus, mobile at 25 °C; it ferments rhamnose but not xylose, is hemolysis-positive and shows a positive result against *Staphylococcus aureus* in the CAMP test [11]. However, as early as 25 years ago, four LM isolates were isolated from red-smear and mold-ripened soft cheeses. These isolates were found to lack hemolysis and/or motility [12] and were thus named atypical LM (aLM). All classical biochemical tests and genetic techniques failed to identify these isolates as *L. monocytogenes* species. Nevertheless, 16rRNA sequence analysis showed them to be 94% homologous to LM, closer than that between *Listeria* spp. and *Brochotrix* spp. In recent years, there have been an increasing number of reports of aLM in various environments showing no or very weak hemolysis but otherwise possessing the biochemical and genetic characteristics typical of *L. monocytogenes* species [13]. Non-hemolytic *L. monocytogenes* are most commonly isolated from food products, but some have also been detected in clinical samples [14].

Despite the rapid development of modern identification methods such as MALDI-TOF and DNA sequencing, these methods are expensive and still not widely available, so phenotypic traits are recommended in ISO methods. Although there are several publications on non-hemolytic LM, none provide many samples together with in-depth analysis of phenotypic traits and data on the correlation between the isolation environment and morphological, biochemical, and genetic traits. Despite the use of *L. monocytogenes* as a model organism, there is great interest in studying these bacteria due to the increasing number of listerioses in the USA and Europe. Therefore, the aim of this study was to isolate non-hemolytic LM from different foods and environments, conduct an in-depth analysis of phenotypic traits, and determine the relationship between isolation environment, morphology, biochemical characteristics, antibiotic sensitivity, and pathogenic potential.

## 2. Materials and Methods

### 2.1. Samples

In Poland, 2495 samples were tested between 2009 and 2019: soil (n = 1000), fruit (n = 160), vegetables (n = 210), ready-to-eat (RTE) food (n = 425), meat (n = 400) and fish raw materials (n = 80), salad ingredients (n = 120), and swabs from the fish/meat processing lines (n = 100) (Table 1). Soil samples were collected in 2010–2018 in two seasons, spring (March–May) and autumn (September–November), from a 250 km^2^ area of the West Pomeranian “Voivodship” in Poland. In selecting soil sampling sites for testing, the criteria that were taken into account were type of land fertilization, degree of human use (determined by interviewing farmers and fruit growers), and the possibility of taking fruit and vegetables from the same sites from which the land was sampled. Table 1 shows the types of soil samples. A sample of 25 g of soil was taken with a corkscrew to a depth of 10 cm and transported to the laboratory in a sterile bag. Fruits and vegetables were collected from the same locations as the soil samples (Table 1) and placed in sterile bags. RTE foods: dumplings (200), croquettes (60), salads (30), sandwiches (10), fish products (45), sprouts (15), sushi (40), desserts (210), carrot (20), carrot juices (20), and ingredients of salads and desserts (263) were purchased from various retail outlets (Table 1). RTE foods and individual salad/dessert ingredients originated from the same batch. The foods and ingredients were supplied by 13 different producers with processing plants located in the West Pomeranian Region of Poland immediately after being manufactured and stored at 4 °C (according to the producer’s recommendation). The detailed breakdown and characterization of 262 LM isolates are provided in RTE food studies [15]. Swabs were taken randomly from different locations within one meat plant (n = 50) and one fish plant (n = 50) (Table 1). A 25 cm^2^ area bound by a template was swabbed with a sterile AMIESA transport swab (Meus, Italy) moistened with sterile 0.85% NaCl and transported to the laboratory at 4 °C [10]. 

### 2.2. Reference Strains

The following were used as reference *L. monocytogenes* strains: 1/2a (ATCC 19111), 1/2b (CIP 7832), 1/2c (ATTC 10112), 3a (ATCC 19113), 3b (CIP 7836), 4a (ATCC 19114), 4b (ATCC 13932), 4c (ATCC 19116), 4d (ATCC 19117), 4e (ATCC 19113), and 7 (NCTC 10890), in addition to *Staphylococcus aureus* (ATCC13565), *Streptococcus pneumoniae* (ATCC 49619), and *Rhodococcus equi* (ATCC 25729). The reference strains were used for each test as a positive control during biochemical identification, serogrouping, the gene virulence test, and the antibiotic resistance test.

### 2.3. Isolation and Identification of L. monocytogenes 

*L. monocytogenes* were isolated and identified from soil, foods, and swabs according to ISO 11290-1 [11]. In addition, a growth assay at 4 °C was performed, and growth on OCLA (Oxoid Chromogenic Listeria Agar, Oxoid, Basingstoke, Hampshire United Kingdon) and RAPID ’L.mono (BioRad, Hercules, CA, USA) was determined. Rapid ‘L. mono medium was used for the identification of *L. monocytogenes* and *Listeria* sp. without confirmation by other methods in accordance with ISO 16140-2:2016 [16]. In addition, OCLA medium is an alternative medium of similar composition to ALOA described in ISO 11290-1 [11]. The motility of LM i aLM was determined on the motility medium (BTL, Łódź, Poland) and the observation of bacterial motility was carried out under a microscope (NIKON, Eclipse E600, Tokyo, Japan) in a drop-hanging preparation [11]. The following biochemical tests were performed: Voges–Proskauer, Methyl-Red, and Nitrate Reduction. Bacterial cell size was measured in a Gram-stained slide under a 0.1 µm light microscope (NIKON, Eclipse E600, Tokyo, Japan). The hemolysis test was performed according to ISO 11290-1:2017 [11]. If the morphological and physiological characteristics were indicative of Listeria spp., blood agar plates (Oxoid with 5% dehydrated sterile sheep blood) were inocualted and incubated for 24 h at 30 °C. A zone of translucence around the colony was considered a positive result. Sugar fermentation was determined on microplates by adding 100 µL of BHI Broth, 20 µL of test sugar solution (Table 4) and 10 µL of bacterial culture, followed by incubation for 5 days at 37 °C [11]. The CAMP test was performed on Blood Agar Base (Oxoid, Basingstoke, Hampshire, United Kingdon) with 5% dehydrated sterile sheep blood (Proanimali, Wrocław, Poland) performed on line cultures according to ISO11290-1 of *S. aureus* (ATCC13565), *R. equi* (ATCC 25729), and aLM test strains [11]. On the basis of the Gram-stained preparation, motility, catalase, and oxidase capacity, *Listeria* sp. genus membership was determined, while hemolysis, CAMP tests, carbohydrate degradability, Voges–Proskauer, Methyl-Red, and Nitrate Reduction determined species-specific classification.

### 2.4. Storage and Revival of Reference LM Strains and aLM Isolates

Reference strains and aLM isolates were stored at −80 °C using a cryobank (Mast Diagnostica GmbH, Reinfeld, Germany). To revive reference strains of *L. monocytogenes*, BHI Broth medium (Scharlau, Barcelona, Spain) was used in which a bead of cryobank was placed, incubated for 48 h at 37 °C, sieved onto LSA medium, and incubated for another 48 h at 37 °C. For the revival of aLM isolates, the modification method described in a patent was used [17]. Briefly, the cryobank bead was placed in BHI Broth medium with 5% defibrinated sterile sheep blood (Proanimali, Wrocław, Poland) and incubated for 48 h at 37 °C, followed by a reduction culture on LSA medium (Oxoid, Basingstoke, Hampshire United Kindon) and incubation for 72 h at 37 °C. Finally, the resultant culture was screened on Blood Agar Base with 5% defibrinated sheep blood and incubated for 48 h at 37 °C, from which biochemical and genetic tests were performed.

### 2.5. DNA Isolation

LM reference bacteria grown 24 h at 37 °C in BHI Broth medium were centrifuged (5 min, 12,000 rpm). DNA was isolated from the bacterial pellet using the Genomic Mini AX Bacteria+ kit (060-60M, A&A Biotechnology, Gdynia, Poland) according to the manufacturer’s protocol. In the case of atypical non-hemolytic LM, an in-house modification described in a patent was used [17]. Briefly, aLM were cultured for 48 h in BHI Broth medium containing 5% defibrinated sterile sheep blood to achieve the higher bacterial precipitate mass required for isolation. During DNA isolation, the hydrolysis step was performed using a combination of available lysozyme, mutanolysin, and proteinase K in a ratio of 10:10:20 at 37 °C, which increased the DNA concentration (ng/µL) and 260/280 purity (Nano Drop ND1000, Thermo Fisher Scientific, Wilmington, DE, USA).

### 2.6. Multiplex PCR Molecular Serotyping

All isolates were classified via multiplex PCR using twelve primers for the genus *Listeria* spp. for the species *L. monocytogenes*. Based on molecular analyses, *L. monocytogenes* were classified into four lineages, with most isolates belonging to lineages I (serotypes 1/2b, 3b,3c, and 4b) and II (serotypes 1/2a, 3a, and 1/2c) [5]. Furthermore, the serotypes were also distinguished into molecular PCR-based serogroups: II.a (with serotypes 1/2a and 3a), IIb (1/2b and 3b), IIC (1/2c and 3c), and IVb (4b, 4d, and 4e) [18]. A list of primers and multiplex PCR reaction conditions is shown in Table 2.

### 2.7. Gene Virulence and Specific Gene for L. monocytogenes

Eleven virulence genes were determined: *hly*, *iap*, *prfA*, *inlA*, *inlB*, *inlC*, *inlJ*, *plcA*, *plcB*, *IIsx*, and *actA*, and 3 specific genes for *L. monocytogenes*: *mpl*, *LMOf 2365_0970*, and *LMOf 2365_2721*. A list of primers and PCR reaction conditions is shown in Table 2.

### 2.8. Electrophoresis of Multiplex PCR and PCR Reaction Products

PCR reaction products obtained during the determination of serotypes and virulence genes having a volume of 8 µL including 6X Loading Buffer TriDye (A&A Biotechnology, Gdynia, Poland) were separated electrophoretically under standard conditions (5 V/cm) on a 2% agarose gel (MAXIMUS, Łódź, Poland). The gel was stained with 4 µL of Midori Green Advance DNA Stain (MG04, Genetics, Düren, Germany) in 1X TBE buffer (10X TBE buffer A3945,1000, A&A Biotechnology, Gdynia, Poland), observed under UV light, and archived (GelDoc, BioRad, Hercules, CA, USA). Product size was compared with the mass marker M100-1000 (MR 65, A&A Biotechnology, Gdynia, Poland).

### 2.9. Antibiotic-Resistant L. monocytogenes

Antibiotic-resistance testing was performed using the Kirby–Bauer method on Mueller–Hinton medium with 5% dehydrated horse blood and 20 mg/L β-NAD + (MH-F Broth) [29,30]. Fourteen antibiotics were used, spanning 10 groups as per the WHO [18] classification: ampicillin (AMP, 10 µg), cephalothin (CET, 30 µg), chloramphenicol (CHL, 30 µg), ciprofloxacin (CIP, 5 µg), clindamycin (CLI, 2 µg), erythromycin (ERY, 15 µg), gentamicin (GEN, 10 µg), kanamycin (KAN, 30 µg), mezlocillin (MEZ, 30 µg), penicillin (PEN, 5 µg), rifampicin (RIF, 5 µg), streptomycin (STR, 25 µg), tetracycline (TET, 30 µg), and vancomycin (VAN, 30 µg). Plates with antibiotic discs were incubated for 18 h at 34 °C. Zones of inhibition of bacterial growth were measured in mm with a ruler (HiAntibiotic ZoneScale, Mumbai, India) and the strains were classified as susceptible, with reduced susceptibility, or resistant, following the criteria of the European Committee on Antimicrobial Susceptibility Testing for *L. monocytogenes* [30].

### 2.10. Statistical Analysis

The reported data are average values from triplicates. The significance of differences was identified using a post hoc Tukey’s test of honestly significant differences (*p* < 0.05). Analyzing so many cases and variables required using several statistical methods. Principal component analysis (PCA) and cluster analysis, e.g., a dendrogram (Ward’s method, Euclidean distances), were used to investigate correlations, as well as to classify samples and morphometric, biochemical, and genetic characteristics. All statistical analyses were performed using Statistica 13.3 (Statsoft, Tulsa, OK, USA).

## 3. Results and Discussion

### 3.1. Isolation Environment of Non-Hemolytic L. monocytogenes

From 2495 samples, 262 (10.5%) *L. monocytogenes* (LM) isolates were isolated, of which 30 (11.5% of 262) were classified as atypical LM (Table 1). Atypical isolates were the most numerous (12 isolates; 40.0%) in samples from land: arable land fertilized with chicken or cattle manure (n = 4), allotment gardens fertilized with manure (n = 2), grassland during intensive grazing of dairy cattle (n = 2), forest hunting perimeter (n = 2), and land in front of a raw material receipt room at a meat plant (n = 2). The season of soil sampling did not affect the occurrence of LM and aLM [31]. These are followed by 11 (36.7%) aLM isolates obtained from vegetables and 3 (10.0%) obtained from fruits that grew in the soils from which LM and aLM were isolated (Table 1). The occurrence of LM in hard-to-reach areas on the production line equipment could be related to biofilm formation, lack of proper hygiene, and the use of an inefficient cleaning and disinfection program in the plant [32,33,34,35]. A further three aLM isolates were isolated from swabs from the production line: the door seal on the production floor, the conveyor belt to the pasteurizer, and the pasteurizer roller. Only one aLM isolate was obtained from RTE food, specifically meat dumplings (Table 1). It is likely that the presence of LM in cultivated soil, vegetables, fruit, and the forest district came from manure and animal droppings [36,37], whereas the presence of LM in the soil in front of the meat plant was associated with the transfer of bacteria on workers’ shoes (data not shown). In a study, Szymczak et al. [15] showed that LM occurred most in croquettes, dumplings, salads, and fish RTE food. Thus, a positive incidence correlation between LM vs. aLM can be seen only for plant foods (organic fertilizers) and meat processing. *L. monocytogenes* is a common threat to the production environment because it is hardly enough to occur in soils with antibiosis of the soil microbiome [38,39], prophage infection [40], and abiotic stress [38,41]. *L. monocytogenes* can also survive in the food production environment for many months and lead to cross-contamination [42] during dumpling production [15] or other processed foods [12]. Despite many publications describing the occurrence of LM in soil [43], processed food, in animals and their habitat [13], and clinical specimens from humans and animals [44,45], there is no information on the occurrence of aLM in soil.

### 3.2. Phenotypic Characteristics

All of the non-hemolytic isolates lacked motility (a generic trait of *Listeria* spp.) in contrast to reference LMs (Table 3). The same pattern was reported in other studies [46,47]. In this study, the lack of phenotypic motility was confirmed in all aLM isolates by the absence of the motility-related gene *flaA* (Similar findings were obtained for aLM isolates from cheese [12]. All tested non-hemolytic LM failed to grow at 4 °C in contrast to the LM reference strains. In contrast, growth at 30 and 37 °C was shown by all reference strains and non-hemolytic LM isolates (Table 3). Similar results were obtained by Orsi and Wiedmann [46], who proposed that *Listeria* bacteria with unique characteristics for movement and growth at different temperatures should be classified into the genus Mesolisteria. The ability to perform β-hemolysis is a defining phenotypic indicator for LM species, which distinguishes them from *L. innocua*. Twenty-five of the aLM isolates (83.3%) were found to lack hemolytic ability (Table 3). These included 12 isolates from the soil, 3 from line swabs, 3 from strawberries, 4 from beetroot, 2 from sprouts, and 1 from meat dumplings. The remaining five aLM isolates had the capacity for hemolysis but no ability to move. Maury et al. [48] also noted that non-hemolytic strains were slower to grow at 37 °C. The lack of hemolytic ability in aLM should not indicate a lack of pathogenic potential, as Lindback et al. [49] showed that non-hemolytic LM introduced at 10^9^ CFU into the liver of mice caused the death of 60% of individuals. *L. monocytogenes* has a *sigB* and *prfA* regulatory system to detect the changing environment and implement survival mechanisms to overcome the diverse, often unfavorable, conditions for this bacterium, but it also has the ability to transition from a harmless saprophyte to a pathogen [50]. Therefore, an additional study should test how the occurrence of non-hemolytic aLM may also be a consequence of *prfA*/*sigB* gene mutations and/or is a result of environmental stress.

In the CAMP test, all 12 aLM isolates from the soil were positive against *S. aureus* despite negative hemolysis (Table 3). In contrast, of the 18 aLM isolates from food and swabs, only 8 showed a positive CAMP result, including 5 isolates capable of hemolysis. Bhalla et al. [44] also obtained negative results from both CAMP and hemolysis tests when investigating clinical LM isolates. On the other hand, there is no information in the literature regarding LM isolates showing both a positive CAMP test and a negative hemolysis test. Results also showed that the identification of non-hemolytic LM on the basis of biochemical indicators (Table 3), which are typical of *L. monocytogenes* species [51,52], provided a false negative result, which is a risk for humans, as these isolates may have pathogenic potential. Mafuna et al. [53] showed that non-pathogenic *L. innocua* isolated from the environment in South Africa developed three or more pathogenicity traits, including CRISPR CAS-type adaptive immune systems. Non-hemolytic LM are most commonly found in food products and food processing environments. In contrast, an LM isolate from pet food was characterized as lacking β-hemolytic isolates and belonged to serotype 1/2a [54].

The average bacterial cell size of the LM reference strains was 2.8 µm (Table 3), which is consistent with the results of Ludwig et al. [52]. For aLM, the cell size ranged from larger (3.0–3.2 µm for the six isolates from manure-fertilized soil) to smaller (2.6 µm for isolates from other soil samples, 2.6 µm from fruit and vegetables, and 2.5 µm from line swabs and 2.0 µm from RTE food) compared to LM reference strains (Table 3). In turn, non-hemolytic aLMs were smaller in size in 18 cases and larger in size than standard LMs in 7 cases. Halter et al. [51] showed that a bacterial cell size from 2.8 to 3.2 µm was characteristic of *L. rocourtiae* and *L. weihenstephanensis*. Most studies on atypical non-hemolytic LM do not include information on bacterial cell size and do not link this characteristic to the capacity for hemolysis. However, the results presented here clearly indicate that the size of aLMs varies, which may be due to the abundance of nutrients in the environment, the growth rate of the bacterial cells, genomic and metabolic improvements, and other factors [55]. Results showed that the largest non-hemolytic aLM cells came from nutrient-rich manure soil, in contrast to the smaller-sized bacteria from non-manured soil or vegetables. It was observed that typical growth of LM reference strains on LSA, RAPID’L.mono, and ALOA media occurred after 48 h of incubation, whereas atypical non-hemolytic LM growth occurred after 72 h, which was probably due to the adaptation of atypical non-hemolytic LM to different environments. Therefore, the method described in the previous section concerning the modification of proliferation with the addition of blood enriched the medium with nutrients and allowed aLM to grow at a similar rate as LM. The hemolytic aLMs had similar biochemical characteristics to the reference LMs, except for growth at 4 °C. In addition, three isolates (B68, KS68, and P54) grew on RAPID’L.mono medium in the form of white colonies typical of *L. innocua* (Table 3).

The dendrogram divided the atypical isolates and reference strains into five groups (A–E) in terms of phenotypic characteristics (Figure 1A). The LM reference strains formed a separate group that was the furthest away (by 21 units) from the other groups of aLM isolates. The most similar phenotypic traits were in groups B and C (only four units different), consisting mainly of isolates from soil, sprouts (RS29), and strawberries (T8). Least dissimilar from the reference strains was group E, representing four isolates from beetroot (B15, B16, B54, and B56), two from strawberries (T16 and T18), one from dumplings (D20), and two from production line swabs (SW1 and SW2).

### 3.3. Ability to Ferment Sugars

*L. monocytogenes* can ferment rhamnose, have no ability to ferment xylose and, depending on the serotype (1/2a, 1/2b, 3b, 4c, and 7), ferment mannitol [11]. LM reference strains were also found to ferment galactose, glucose, lactose, methyl-L-D-glucoside, and sucrose, but not D-arabitol, arabinose, glucose-1-phosphate, ribose, sorbitol, D-tagatose, and D-xylose (Table 4). Thus, LM reference strains were able to ferment 6–7 of the 14 sugars tested simultaneously, while atypical non-hemolytic LM fermented between 5 and 11 sugars simultaneously. Each of the 14 sugars tested was degradable by at least one aLM, which was not the case for the LM reference. All atypical non-hemolytic LM isolates fermented D-arabitol and rhamnose (Table 4). Methyl-L-D-glucoside was fermented by 28 isolates; ribose, glucose, and glucose-1-phosphate by 17–20 isolates; lactose and sucrose by 12 isolates; D-tagatose and D-mannitol by 3–4 isolates; galactose, arabinose, and sorbitol by only 1–2 isolates (Table 4). Isolates 111 (11 sugars), SW1 (9), 281, 288, and 140 (8 sugars) fermented the most types of sugars, in contrast to isolates 112, B67, B68, CA11, and SW3, which fermented only five sugars (Table 4). The two atypical LM isolates (11 and SW1) that fermented the most types of sugars were also able to degrade two of the three least-fermented sugars. The isolates from soil had the richest amylase apparatus, while the other isolates, mainly those from vegetables, had the most modest fermentation (energy acquisition) ability. The results confirm that *Listeria*, especially from soil, can utilize multiple carbon sources (a trait attributable to its large number of carbohydrate transporters), which may influence virulence gene expression [56]. It was observed that aLM isolates gained the ability to ferment additional sugars mainly at the expense of the ability to ferment galactose and/or sucrose by sacrificing galactose and/or sucrose fermentability to a level inferior to the reference LM (Table 4). Gasanov et al. [57] reported that the identification of non-hemolytic LM should be based on differences in the fermentation of certain sugars, even though biochemical identification markers are sometimes difficult to interpret as color reactions can be ambiguous. On the other hand, Jarvis et al. [58] proposed that aLM-fermented sugars should be used as a component of the culture medium, which promotes and improves the growth of aLM while increasing the detection of LM.

**Table 4 foods-12-03630-t004:** Sugar fermentation ability by reference strains and atypical *L. monocytogenes* isolates.

Sample	D-arabitol	Arabinose	Galactose	Glucose-1-Phosphate	Glucose	Lactose	D- Mannitol	Methyl-L-D-Glucoside	Rhamnose	Ribose	Sorbitol	Sucrose	D -Tagatose	D-Xylose
1/2a	−	−	+	−	+	+	+	+	+	−	−	+	−	−
1/2b	−	−	+	−	+	+	+	+	+	−	−	+	−	−
1/2c	−	−	+	−	+	+	−	+	+	−	−	+	−	−
3a	−	−	+	−	+	+	−	+	+	−	−	+	−	−
3b	−	−	+	−	+	+	+	+	+	−	−	+	−	−
3c	−	−	+	−	+	+	−	+	+	−	−	+	−	−
4a	−	−	+	−	+	+	−	+	+	−	−	+	−	−
4b	−	−	+	−	+	+	−	+	+	−	−	+	−	−
4c	−	−	+	−	+	+	+	+	+	−	−	+	−	−
4d	−	−	+	−	+	+	−	+	+	−	−	+	−	−
4e	−	−	+	−	+	+	−	+	+	−	−	+	−	−
7	−	−	+	−	+	+	+	+	+	−	−	+	−	−
112	+	−	−	−	−	+	−	+	+	−	−	+	−	−
245	+	−	−	−	−	+	−	+	+	−	−	+	−	+
273	+	−	−	−	−	+	+	−	+	+	−	+	−	−
275	+	−	−	−	−	+	−	+	+	−	+	+	−	−
280	+	−	−	−	+	−	+	−	+	+	−	+	−	−
281	+	−	−	+	+	−	−	+	+	+	−	+	−	+
288	+	−	−	+	+	−	−	+	+	+	−	+	−	+
324	+	−	−	−	−	+	−	+	+	−	−	+	−	+
352	+	−	−	−	+	−	−	+	+	+	−	+	−	+
69	+	−	+	−	−	+	−	+	+	−	−	+	−	−
111	+	+	−	+	−	+	+	+	+	+	−	+	+	+
140	+	−	−	+	+	−	−	+	+	+	−	+	−	+
B15	+	−	−	+	+	−	−	+	+	+	−	−	−	−
B54	+	−	−	+	+	−	−	+	+	+	−	−	−	−
B56	+	−	−	+	+	−	−	+	+	+	−	−	−	−
B16	+	−	−	+	+	−	−	+	+	+	−	−	−	−
RS26	+	−	−	+	+	−	−	+	+	+	−	−	−	−
RS29	+	−	−	+	+	−	−	+	+	+	−	−	−	−
B67	+	−	−	−	−	+	−	+	+	−	−	−	−	+
B68	+	−	−	−	−	+	−	+	+	−	−	−	−	+
D20	+	−	−	+	+	−	−	+	+	+	−	−	−	−
KS68	+	−	−	+	+	−	−	+	+	+	−	−	−	−
T8	+	−	−	+	+	−	−	+	+	+	−	−	−	−
T16	+	−	−	−	−	+	+	+	+	−	+	−	−	+
T18	+	−	−	+	+	−	−	+	+	+	−	−	−	−
CA11	+	−	−	−	−	+	−	+	+	−	−	−	−	+
P54	+	−	−	+	+	−	−	+	+	+	−	−	−	−
SW1	+	+	−	+	−	+	−	+	+	+	−	−	+	+
SW3	+	−	−	−	+	−	−	+	+	−	−	−	+	−
SW10	+	−	−	+	+	−	−	+	+	+	−	−	−	−

“+”—presence of the tested trait; “−“—no survey trait.

Five aLM isolates (B67, B68, KS68, CA11, and P54) with the characteristics of *L. innocua* on chromogenic media were hemolyso-positive, not motile, rhamnose fermentable, xylose-positive (B67, B68, and CA11), mannitol-negative, CAMP-*S. aureus*-positive, CAMP-*R. equi*-negative, and had one gene belonging to the LIPI-1 pathogenicity island and the LM-specific gene LMOf2365_2721 (Table 3, Table 4 and Table 5). Johnson et al. [59] found that the atypical *L. innocua* isolate was also hemolyso-positive but motile; non-fermentation of rhamnose, xylose, and mannitol was also observed, along with six genes belonging to the LIPI-1 pathogenicity island. Additionally, Moura et al. [60] described atypical *L. innocua* hemolyso-positive isolates that had six genes belonging to the LIPI-1 pathogenicity island, but the isolates were motile and did not show data regarding the fermentation of sugars. As a result of differences from other authors, these five atypical LMs cannot be classified as atypical *L. innocua*.

Statistical grouping divided the reference strains and isolates into four groups (A–D). The reference strains were a separate group A and were the most different from group D (23 units away), which consisted mainly of vegetable-borne isolates (beetroot, kale sprout, radish, and meat dumpling isolates), whose fermentation capacity was limited to D-arabitol, glucose, glucose-1-phosphate, methyl-L-D-glucoside, rhamnose, and ribose (Figure 1B). The reference strains were 17 units away from groups B and C, which were the most similar to each other (only seven units apart). Group B consisted of isolates from soil, vegetables, and fruit, while group C consisted of isolates from soil and the meat plant and were able to ferment the most diverse array of sugars.

### 3.4. Antibiotic Susceptibility of aLM

LM reference strains, especially 1/2a 1/2b 1/2c, 3a, and 4a, were sensitive to all 14 antibiotics, especially AMP10, ERY15, RIF5, TET30, and PEN10, while CLI2 had the least-pronounced effect on these strains (14.7 mm) (Table 5). The low value of the standard deviation (SD = 2–3 mm) indicates that the antibiotics tested had a similar effect on the reference strains, except for AMP10, PEN10, MEZ30, and ERY15. For aLM isolates, 11 cases (2.6%) of antibiotic resistance were found across the 420 tests: five isolates on CLI2, four isolates on TET 30, and one isolate each on STR25 and PEN10 (Table 5). Resistance to two antibiotics was exhibited by isolates SW3 and KS68, and to one antibiotic by isolates D20, RS26, RS29, B15, 111, 69, and 280. These included isolates from the soil, beet, sprout, and swab. There is no information in the literature on the prevalence of antibiotic-resistant atypical L. monocytogenes. The average antibiotic resistance of aLM was 23.6 mm, lower than the average for the reference strains (25.7 mm). The sensitivity of atypical isolates was the highest to MEZ30, ERY15, AMP10, CHL30, and GEN10 in contrast to the antibiotic CLI2 (14.9 mm). The most variable resistance of atypical isolates was to the antibiotics TET30, CLI2, PEN10, STR25, AMP10, and KAN30, while VAN30, ERY15, and GEN10 had the most consistent effect. There was no significant correlation between aLM antibiotic resistance and the ability to hemolyse, except for a weak correlation (0.387) between non-hemolysis and lowered sensitivity to MEZ30.

Gomez et al. [61] and Osaili et al. [62] showed that 35% of LM isolates isolated from poultry, pork, and beef and 56.4% of isolates from various cheese species were resistant to clindamycin. Thedieck et al. [63] suggested that the increased antibiotic resistance in *L. monocytogenes* is due to the activation of the antimicrobial peptide sensor system (as the bacterium detects increases in antibiotic concentrations) and that active efflux systems may contribute to *Listeria*’s adaptation to changing environmental factors. Anthropogenic factors are important in the emergence of new antibiotic resistance in aLM capable of spreading in nature and consequently into food production and food environments [10]. Therefore, continuous monitoring of antibiotic resistance, including aLM isolates, is important due to the potential emergence of CRISPR CAS-type adaptive immune systems [64]. Gentamicin resistance was demonstrated in 4.6% of LM isolates from food, food processing plants, and sick humans in Germany [65] and in 1.2% of isolates from food in China [66]. The possibility of LM acquiring antibiotic resistance may result from the selection of the species with intrinsic or acquired resistance to antibiotics, biocides, and heavy metals in soil and groundwater, as well as natural tolerance to extreme conditions [10].

The dendrogram divided the samples into six groups labeled A–F, with reference strains relegated to A–C (Figure 1C). Groups A and B were the least distant from each other by 10 units, where LM strains were most sensitive to ERY15, AMP10, and MEZ30 and least sensitive to CLI2 and STR25. Group A, in addition to the LM reference strains, also included the three isolates 273, 275, and 352. Group C was 13 units away from groups A–B and consisted of only two samples (3a LM and B68 aLM), which were most sensitive to ERY15, AMP10, MEZ30, and TET30, while most resistant to CLI2 and CIP5. The next groups, D–F, were 17 and 26 units away from A–C, respectively. The sensitivity of groups D–F to some antibiotics (AMP10 and/or MEZ30) was the same as in groups A–C, but high efficacy turned into low sensitivity for the other antibiotics.

### 3.5. Gene Virulence and Serological Classification

The genes *hly*, *prfA*, and *iap* were present in all LM references, except *iap* in strains 4a and 4c (Table 6). In contrast, all atypical non-hemolytic LMs, irrespective of origin, possessed the *hly* and *prfA* genes, while none possessed the *iap* gene. LM references showed 10–15 of the virulence genes tested, with serotypes 4a and 4c having the fewest. The atypical non-hemolytic LM isolates possessed only 3–7 virulence genes, with the highest number (6–7 genes) possessed by isolates 273, 288, and B54 from soil, mainly manured or located in close proximity to a meat processing plant, similar to the findings of Soni et al. [67]. Only three virulence genes were present in 15 isolates, including those from soil (5), vegetables (4), fruit (2), dumplings (1), and line swabs (3). Genes encoding internalin (*inlA*, *inlB*, *inlC*, and *inlJ*) were not present in any aLM, in contrast to the reference LMs (Table 6). This may suggest that aLMs have a lower virulence capacity during intestinal infection, entry into host cells, and adaptation to the intracellular lifestyle [60]. Maury et al. [48] showed that LMs without the capacity for hemolytic activity are rare (0.1%), tend to be hypo- rather than hypervirulent clones, and have reduced virulence. It was observed that hypovirulent LM clones have greater adaptability to food processing environments, having more genes involved in stress resistance and disinfectant tolerance in exchange for fewer virulence genes, in contrast to hypervirulent clones [68]. This may explain in this study the greater diversity of phenotypic traits in atypical non-hemolytic LMs with fewer virulence genes in contrast to reference LMs (Table 6). Maury et al. [68] showed that atypical *L. innocua* might represent an intermediate evolutionary stage with subsequent loss of virulence genes leading to niche restriction similar to non-hemolytic LM. A trend of gene loss was also demonstrated within the LIPI3 and LIPI-4 regions [59,60]. Quereda et al. [69] concurred that the acquisition and loss of genetic elements provide the traits necessary for LM specialization in an environment or host. It is also possible that the repeated use of only a dozen reference strains in many studies has led to an underestimation of the overall LM biodiversity so that any dissimilarity is now considered atypical. Furthermore, there is still no designated species or typical gene to guarantee the correct identification of LM species.

Of the 30 aLM isolates, only five were classified into a serogroup: isolates 111, 140, and 352 into serogroup 4a–4c, and isolates 273 and 288 into serogroup ½c–3c (Table 6). The remaining aLM were not classified into any serogroup due to the absence of genes *ORF2819*, *ORF2110*, *lmo0737*, and *lmo118*. Feng et al. [70] showed that atypical non-rhamnose-fermenting LMs belonging to serotype 4h are not identified by PCR serotyping techniques. The present study suggests otherwise, as all atypical non-hemolytic LM isolates were found to ferment rhamnose. For aLM, the *prfA* gene correlated strongly with sucrose fermentability (0.612, *p* = 0.000) and weakly with galactose fermentability (0.371, *p* = 0.043), while the *mpl* gene correlated strongly with sucrose fermentability (0.480, *p* = 0.007). The *prfA* gene correlated strongly with the presence of the *mpl* gene (0.785, *p* = 0.000) and moderately with the *lmo1118* gene (0.447, *p* = 0.013) present in serogroup 1/2c-3c. The hemolytic capacity of aLM correlated more strongly with the *ORF2819* gene (0.599, *p* = 0.000) than with *ORF2110* (0.415, *p* = 0.023). In addition, 25 out of 30 aLM isolates had the *hly* gene but no hemolytic ability and no *plcA* or *inlC* genes. Similarly, Moreno et al. [12] isolated aLM from meat and production environments that contained the *inlC*, *hly*, and *plcA* genes, although they showed very weak or no hemolysis.

Out of the 29 known *Listeria* spp., *L. immobilis* showed the most similar characteristics to the aLM tested. However, *L. immobilis* is simultaneously characterized by a lack of motility and hemolysis. Furthermore, as opposed to the tested aLMs, *L. immobilis* does not grow on BHI Broth at 4 °C, does not ferment L-rhamnose, lacks the ability for PI-PLC activity, and lacks the *prfA*, *hly*, and LM-specific gene: *LMOf2365_2721*. Therefore, despite many similarities, the aLM isolates in this study cannot be classified into the species *L. immobilis*.

Statistical grouping divided the reference strains and atypical isolates into four groups, A–D (Figure 1D). Group A (with reference strains) differed by 34 units from groups B–D (with aLM isolates). The largest group, B, which included no less than 15 isolates, had three virulence genes. Group C was the least different from group A and consisted of eight isolates (mainly from the soil) possessing 4–7 genes, including the *mpl* gene. The smallest group, D, contained seven aLM isolates, including four of the five non-hemolytic isolates that possessed four genes each. The most similar aLM isolates were in groups C and D. The presence of the *LMOf2365_2721* gene responsible for encoding glycosyl hydrolase, which is a key enzyme for carbohydrate metabolism, was detected in all model LM and aLMs, while only in group B, represented by isolates from soil (273, 288, 352, and 140) and beetroot (B54), was the *LMOf 2365_0970* gene (encoding a conserved hypothetical proteinase) confirmed. These genes can be used to detect LM in food and are more specific than *hly* or *prfA* genes [24].

### 3.6. PCA Correlation Analysis

Two correlation analyses were performed, the first of which showed the relationship between the biochemical traits of LM strains and aLM isolates (Figure 2), while the second correlation showed the relationships between all LM and aLM samples tested based on their biochemical and genetic traits (Figure 3). 

For the former, the first two principal components (PCs) accounted for 65.33% of the total variance. Many interesting correlations were found (Figure 2), including the ability to move with the ability to grow at 4 °C, the fermentation of galactose, and the occurrence of the genes *actA*, *plcA*, *inlJ*, *plcB*, and *inlA*, showing a very strong positive correlation.

The capacity for hemolysis correlates very strongly and positively with *prfA*, *inlB*, and *mpl* genes but strongly and negatively with D-arabitol fermentation. Larger LM cell size shows a positive correlation with greater sensitivity to most antibiotics (especially MEZ30, CET30, AMP10, and PEN10), a negative correlation with CIP5, and no correlation with STR25. On the other hand, the correlations of cell size vs. antibiotics were average, indicating that antibiotic sensitivity also depends on other traits, e.g., the presence of virulence genes, which also correlated on average (max 50%) with antibiotic sensitivity. PCA analysis showed several correlations between the fermentability of different sugars, with positive correlations being sucrose vs. lactose, sorbitol vs. xylose, and ribose vs. glucose-1-phosphate. Negative correlations were found in lactose vs. ribose, galactose vs. D-arabitol, and xylose vs. glucose. There was a positive correlation between LM sensitivity to the antibiotic CIP5 and glucose-1-phosphate fermentability, as well as correlations of RIF5 and TET30 vs. lactose and sucrose. The CAMP test correlated positively with sucrose and lactose fermentability and with TET30, ERY15, and RIF5 antibiotic sensitivity, while it correlated negatively with glucose-1-phosphate, ribose, and CIP5.

PCA analysis divided the references and isolates into two separate oval clusters that align along the vertical axis with a weak slope to the horizontal axis, the slope being for aLM rather than LM (Figure 3). The first two principal components (PCs) accounted for 75.33% of the total variance. The apparent separate alignment of the LM and aLM strains was probably caused mainly by differences in virulence genes and in morphological and biochemical traits (Figure 1A,D), and to a lesser extent by differences in sugar fermentability and antibiotic sensitivity (Figure 1B,C). The group of standards spanned 4.77 units, while the group of isolates spanned 8.15 units (Figure 3), confirming that the group of isolates has more diverse biochemical and genetic characteristics than the group of reference strains. The standards formed three groups (A, B, and C) corresponding to the phylogenetic groups, with the exception of strains 3a and ½b. The isolates also formed three groups (D, E, and F), of which D contained five isolates from strawberry T16, beetroot B68, carrot CA11, and from cattle intensive-grazing meadows (273 and 275). Group E had the largest, with as many as 17 isolates from soil (9 isolates), vegetables (5 isolates), fruit (2 isolates), and swabs (1 isolate). Group F contained eight isolates from kale (KS 68) and radish sprouts (RS 26 and RS29), beetroot (B15 and B56), meat dumplings (D20), and lineage swabs (SW1 and SW3). The analysis showed that reference strains 4a and 4c were most similar to aLM isolates from group E, especially to isolate 69. Among the aLM isolates, isolates 273 and B68 were the most distantly related to B15, RS29, and D20, mainly based on the CAMP test.

Thus, statistical analysis confirmed that tested aLM isolates differ significantly from the reference LM. The greatest differences between LM and aLM are sequentially due to the number of virulence genes, phenotypic characteristics (hemolysis, motility, cell size, etc.), ability to ferment sugars, and antibiotic sensitivity. All aLM are distinguished from LM reference by their lack of hemolysis, motility, growth at 4 °C, ability to ferment D-arabitol, and lack of six specific genes (*iap*, *inlA*, *inlJ*, *plcA*, *plcB*, and *actA*), while for the remaining phenotypic traits, further differences depend on the aLM isolate. For example, aLM isolates in group D (Figure 3) also differ from the reference LM in having a larger cell size, lacking glucose fermentation, and the ability to ferment sorbitol and xylose. In contrast, aLM isolates in group E (Figure 3) differ from LM additionally in the absence of galactose, lactose, and sucralose fermentation, glucose-1-phosphate fermentation, and ribose fermentation. Isolates from soil in group E are less sensitive to antibiotics, especially TET30 and AM10, while isolates from food can be more sensitive to antibiotics, especially CLI2 and CIP5. In contrast, in the case of group F, aLM isolates also differ from LM in having a smaller cell size, a negative CAMP test of *S. aureus*, and less sensitivity to antibiotics, especially TET30, PEN10, AM10, KAN30, and RIF5. Therefore, phenotypic analysis in the case of aLM is fingerprinting and also allows differentiation of individual isolates.

## 4. Conclusions

Atypical non-hemolytic LM isolates accounted for more than 10% of LM samples from food production environments. The environment influenced both the frequency of occurrence of atypical non-hemolytic LM and their unique phenotypic traits, indicating that animal/plant farming and food processing may be responsible for the proliferation of atypical non-hemolytic LMs. Atypical LM isolates are characterized by greater diversity (broader scope) in terms of morphology and biochemical characteristics but several times lower number of virulence genes compared with the LM reference strains. These basic phenotypic traits make it possible to distinguish aLM from LM, *L. immobilis*, *and L. innocua*, as well as to differentiate between individual aLM isolates. Therefore, in the case of aLM, phenotypic methods constitute fingerprinting. The results showed that the primary distinguishing trait of aLM is the lack of hemolysis, which was very strongly and positively correlated with the *prfA*, *inlB*, and *mpl* genes and strongly negatively correlated with D-arabitol fermentation. According to the three characteristics tested, atypical LM can be divided into three groups: (a) hemolyso-positive and CAMP test against *S. aureus* positive, (b) hemolyso-negative and CAMP test against *S. aureus* positive, and (c) hemolyso-negative and CAMP test against *S. aureus* negative. This study showed a more than 10-fold increase in the prevalence of non-hemolytic LMs compared to other authors’ data. The increase in the incidence of atypical non-hemolytic LM may pose a risk to humans, as they may not be detected by reference methods and have greater antibiotic resistance than LM, even to clindamycin, penicillin, streptomycin, and tetracycline. The standard according to which the isolation and identification of *L. monocytogenes* is performed should be supplemented by the identification of non-hemolytic LM isolates that may pose a public health risk. The discovered atypical non-hemolytic LM isolates require further study, especially using MALDI-TOF and DNA sequencing and possibly transmitting resistance and pathogenicity traits to other *Listeria monocytogenes* cells.

## Figures and Tables

**Figure 1 foods-12-03630-f001:**
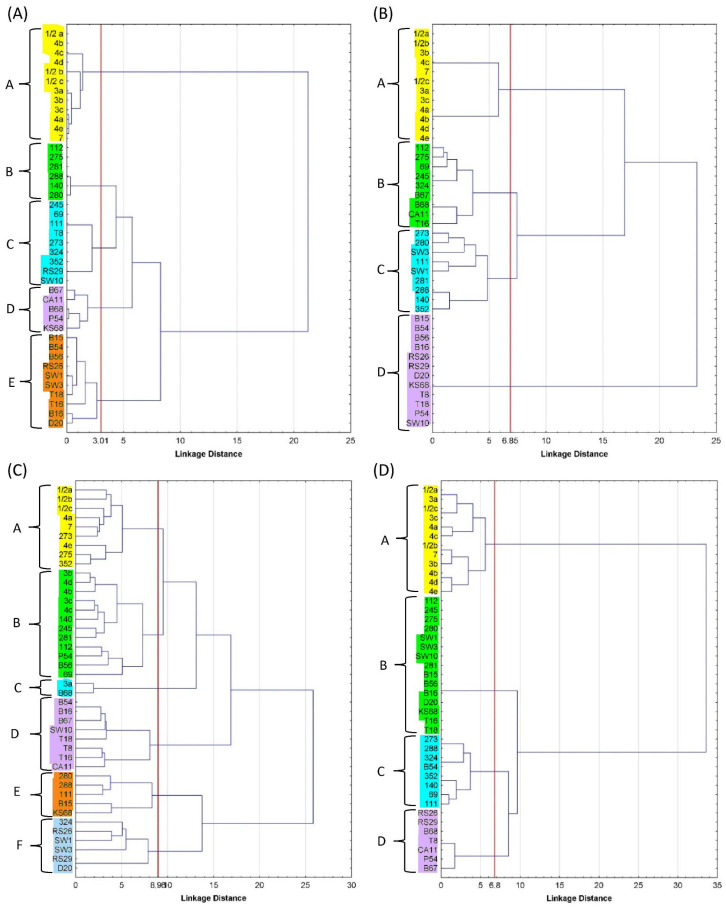
Dendrogram of reference LMs and atypical LMs, grouped by (**A**) selected morphological and biochemical characteristics, (**B**) ability to ferment sugars, (**C**) antibiotic resistance, and (**D**) presence of virulence genes. Groups of isolates highlighted in different colours on the dendrograms are indicated by the letters A–F.

**Figure 2 foods-12-03630-f002:**
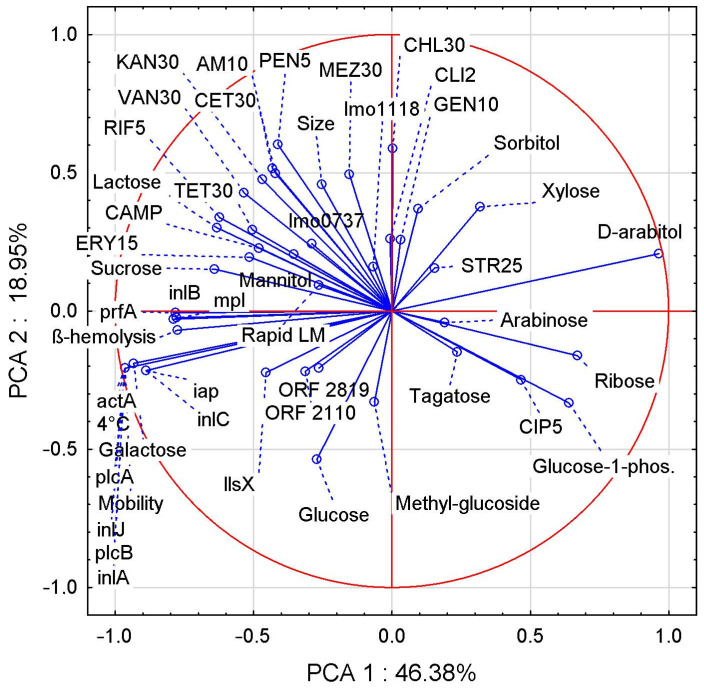
PCA correlation between biochemical attributes evaluated for LM reference strains and aLM isolates.

**Figure 3 foods-12-03630-f003:**
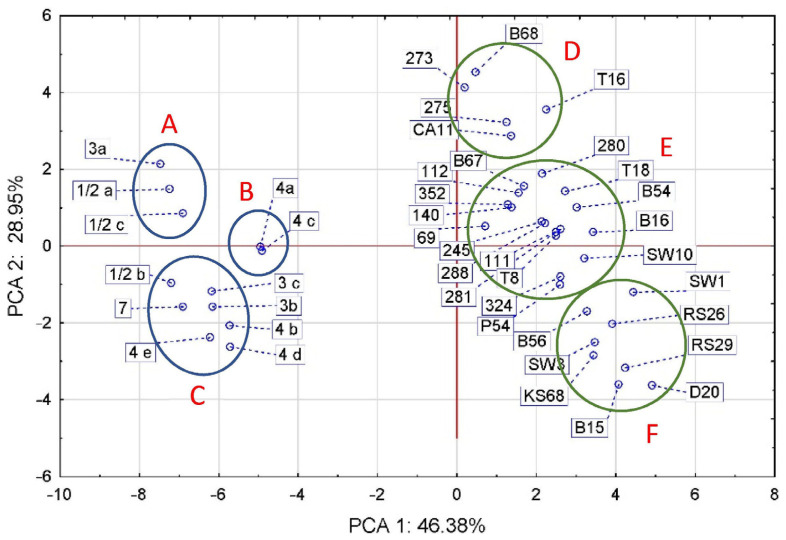
PCA correlation between LM references and aLM isolates based on the analyzed phenotypic traits and genes. The groups LM references are indicated by A–C letters and groups aLM isolates by D–F letters.

**Table 1 foods-12-03630-t001:** Sampling locations and origin of atypical non-hemolytic *L. monocytogenes* (aLM) isolates.

Environmental	Sampling Locations	Number of Analyzed Samples	Count of LM Isolates	Count of aLM Isolates	aLM Isolate Number
Soil (S)	arable soil with natural fertilization (S1)	180	7	4	245, 280, 281, 288
	arable soil artificially fertilized (S2)	180			
	waste soil	180			
	garden plots (S3)	180	8	2	324, 112
	orchards	30			
	meadows	35	4		
	intensive grazing of cattle	50	18	2	273, 275
	usable meadows	50			
	forest (S4)	79	20	2	352, 69
	area around meat processing plant	36	9	2	111, 140
Fruit	strawberry from S1	20	3	1	T8
	strawberry from S2	20			
	strawberry from S3	20	5	2	T16, T18
	raspberry from S2	20			
	raspberry from S3	20			
	blackberry from S4	40			
	blueberry from S3	20			
Vegetables	beetroot from S1	10	5	4	B15, B16, B67, B68
	beetroot from S2	10			
	beetroot from S3	10	4	2	B54, B56
	cabbage from S1	10			
	cabbage from S2	10			
	cabbage from S3	10			
	carrot from S1	10	3	1	CA11
	carrot from S2	10			
	carrot from S3	10	2		
	lettuce from S1	10			
	lettuce from S2	10			
	lettuce from S3	10			
	parsley from S1	10	2		
	parsley from S2	10			
	parsley from S3	10	1		
	potato from S1	10	3		
	potato from S2	10			
	potato from S3	10	2	1	P54
	tomato from S1	10			
	tomato from S2	10			
	tomato from S3	10			
RTE food	dumplings	200	41	1	D20
	croquettes	60	14		
	salads	40	14		
	sandwiches	40	2		
	sushi	40	8		
	salted fish	5			
	smoked fish	5	1		
	fish paste	5	2		
	cold marinated fish	30	4		
	desserts	210			
	vegetable juice	40	2		
Meat raw materials	pork-shoulder	30	10		
	neck of pork	30	5		
	pork ham	30	8		
	intestines	30	12		
	pork loin	30	7		
Fish raw materials	Norwegian salmon	40	10		
	trout	20			
	tuna	20			
Salads ingredients	iceberg lettuce	20			
	kale sprouts	20	5	1	KS68
	radish sprouts	20	6	2	RS26, RS29
	radish	20	3		
	tomato	20			
	feta cheese	20			
Swabs in fish plants	tunnel for the pasteurizer	5			
	production line floor	5	1		
	water drainage system	5			
	pasteurizer turbine	5			
	seal in the hall door	5	3	1	SW10
	floor drain	5	2		
	transport box	5	1		
	pasteurizer	5	2		
	knife	5			
	floor in the raw material cold store	5	1		
Swabs in meat plants	pasteurizer belt	5	2	1	SW1
	pasteurizer rollers	5	1	1	SW3
	seals for cold store doors	5	1		
	sluice gaskets for the production hall	5	2		
	knife	5			
	shelf for storing knives after sterilization	5			
	floor drain	5	1		
	production line floor	5	1		
	transport box for raw material	5	2		
	product transport box	5	0		
All samples	total:	2495	262	30	

**Table 2 foods-12-03630-t002:** Primers used for *L. monocytogenes* virulence genes.

Target Gene	Size of Amplified Product (bp)	Primer Sequences (5′–3′)	PCR Conditions	Reference
*prfA*	274	F: GATACAGAAACATCGGTTGGC	94 °C—3 min; 36 cycles: 94 °C—40 s, 60 °C—45 s, 72 °C—1,15 min; 72 °C—7 min	[19]
		R: GTGTAATCTTGATGCCATCAGG		
*hly*	456	F: GCAGTTGCAAGCGCTTGGAGTGAA	95 °C—2 min; 35 cycles: 95 °C—15 s, 59,5 °C—30 s, 72 °C—1,30 min; 72 °C—10 min	[20]
		R: GCAACGTATCCTCCAGAGTGATCG		
*iap*	131	F: ACAAGCTGCACCTGTTGCAG	95 °C—2 min; 35 cycles: 95 °C—15 s, 58 °C—30 s, 72 °C—1,30 min; 72 °C—10 min	[21]
		R: TGACAGCGTGTGTAGTAGCA		
*plcA*	326	F: CTCGGACCATTGTAGTCATCTT	95 °C—2 min; 35 cycles: 95 °C—15 s, 62 °C—30 s, 72 °C—1,30 min; 72 °C—10 min	[22]
		R: CACTTTCAGGCGTATTAGAAACGA		
*plcB*	261	F: GGG AAA TTT GAC ACA GCG TT	94 °C—3 min; 35 cycles: 94 °C—1 min, 62 °C—2 min, 72 °C—1 min; 72 °C—7 min	[23]
		R: ATT TTC GGG TAG TCC GCT TT		
*mpl*	679	F: TGATGAAATAAAGGTCCACG	94 °C—3 min; 35 cycles: 94 °C—30 s, 60 °C—30 s, 72 °C—40 s; 72 °C—10 min	[24]
		R: CAAGCCATAATGAACAAACG		
*actA*	827	F: GCTGATTTAAGAGATAGAGGAACA	95 °C—2 min; 40 cycles: 95 °C—10 s, 60 °C—30 s, 72 °C—30 s; 72 °C—10 min	[25]
		R: TTTATGTGGTTATTTGCTGTC		
*inlA*	800	F: ACGAGTAACGGGACAAATGC	94 °C—2 min; 30 cycles: 94 °C—20 s, 60 °C—20 s, 72 °C—50 s; 72 °C—7 min	[26]
		R: CCCGACAGTGGTGCTAGATT		
*inlB*	884	F: TGGGAGAGTAACCCAACCAC	94 °C—2 min; 30 cycles: 94 °C—20 s, 65 °C—20 s, 72 °C—50 s; 72 °C—7 min	
		R: GTTGACCTTCGATGGTTGCT		
*inlC*	517	F: AATTCCCACAGGACACAACC	94 °C—2 min; 30 cycles: 94 °C—20 s, 60 °C—20 s, 72 °C—50 s; 72 °C—7 min	
		R: CGGGAATGCAATTTTTCACTA		
*inlJ*	238	F: TGTAACCCCGCTTACACAGTT	94 °C—2 min; 30 cycles: 94 °C—20 s, 60 °C—20 s, 72 °C—50 s; 72 °C—7 min	
		R: AGCGGCTTGGCAGTCTAATA		
*llsX*	200	F: TTATTGCATCAATTGTTCTAGGG	95 °C—3 min; 45 cycles: 95 °C—30 s, 60 °C—1 min, 72 °C—1 min; 72 °C—10 min	[27]
		R: CCCCTATAAACATCATGCTAGTG		
*prs*	370	F: GCTGAAGAGATTGCGAAAGAAG	94 °C—3 min, 35 cycle: 94 °C—40s, 53 °C—1.15 min, 72 °C—1.15 min;72 °C—7 min	[18]
		R: CAAAGAAACCTTGGATTTGCGG		
*ORF2819*	471	F: AGCAAAATGCCAAAACTCGT		
		R: CATCACTAAAGCCTCCCATTG		
*ORF2110*	597	F: AGTGGACAATTGATTGGTGAA		
		R: CATCCATCCCTTACTTTGGAC		
*lmo0737*	691	F: AGGGCTTCAAGGACTTACCC		
		R: ACGATTTCTGCTTGCCATTC		
*lmo1118*	906	F: AGGGGTCTTAAATCCTGGAA		
		R: CGGCTTGTTCGGCATACTTA		
*LMOf2365_0970*	386	F: GCTCAGCGGCAAATCAAAC	94 °C—3 min; 35 cycles: 94 °C—30 s, 60 °C—30 s, 72 °C—40 s; 72 °C—10 min	[24]
		R: GGCACTCGCAACAGAAACG		
*LMOf2365_2721*	583	F: GTTCGTCGGTCCGTGGTA	94 °C—3 min; 35 cycles: 94 °C—30 s, 60 °C—30 s, 72 °C—40 s; 72 °C—10 min	[28]
		R: TTGGCAAGCAAGCAGTTCA		

**Table 3 foods-12-03630-t003:** Phenotypic and biochemical identification of *L. monocytogenes* reference strains and atypical isolates.

Sample	Cell Size	Mobility	Growth at (°C)		β—Hemolysis	CAMP Test		Chromogenic Medium		Esculin, Voges–Proskauer, Methyl Red	Nitrate Reduction
			4	30 and 37		*S. aureus*	*R. equi*	OCLA and ALOA	Rapid L. mono		
1/2a	3.0	+	+	+	+	+	−	BH	GB	+	−
1/2b	2.0	+	+	+	+	+	−	BH	GB	+	−
1/2c	2.8	+	+	+	+	+	−	BH	GB	+	−
3a	2.8	+	+	+	+	+	−	BH	GB	+	−
3b	2.8	+	+	+	+	+	−	BH	GB	+	−
3c	2.7	+	+	+	+	+	−	BH	GB	+	−
4a	2.7	+	+	+	+	+	−	BH	GB	+	−
4b	3.0	+	+	+	+	+	−	BH	GB	+	−
4c	3.0	+	+	+	+	+	−	BH	GB	+	−
4d	3.0	+	+	+	+	+	−	BH	GB	+	−
4e	2.6	+	+	+	+	+	−	BH	GB	+	−
7	2.6	+	+	+	+	+	−	BH	GB	+	−
112	3.0	−	−	+	−	+	−	BH	GB	+	−
245	2.0	−	−	+	−	+	−	BH	GB	+	−
273	2.5	−	−	+	−	+	−	BH	GB	+	−
275	3.0	−	−	+	−	+	−	BH	GB	+	−
280	3.2	−	−	+	−	+	−	BH	GB	+	−
281	3.0	−	−	+	−	+	−	BH	GB	+	−
288	3.0	−	−	+	−	+	−	BH	GB	+	−
324	2.5	−	−	+	−	+	−	BH	GB	+	−
352	2.5	−	−	+	−	+	−	BH	GB	+	−
69	2.0	−	−	+	−	+	−	BH	GB	+	−
111	2.0	−	−	+	−	+	−	BH	GB	+	−
140	3.0	−	−	+	−	+	−	BH	GB	+	−
B15	2.0	−	−	+	−	−	−	BH	GB	+	−
B54	2.5	−	−	+	−	−	−	BH	GB	+	−
B56	2.5	−	−	+	−	−	−	BH	GB	+	−
B16	2.5	−	−	+	−	−	−	BH	W	+	−
RS26	2.5	−	−	+	−	−	−	BH	GB	+	−
RS29	2.5	−	−	+	−	+	−	BH	GB	+	−
B67	2.5	−	−	+	+	+	−	BH	GB	+	−
B68	3.0	−	−	+	+	+	−	BH	W	+	−
D20	2.0	−	−	+	−	−	−	BH	W	+	−
KS68	2.0	−	−	+	+	+	−	BH	W	+	−
T8	2.0	−	−	+	−	+	−	BH	GB	+	−
T16	3.5	−	−	+	−	−	−	BH	GB	+	−
T18	2.8	−	−	+	−	−	−	BH	GB	+	−
CA11	3.2	−	−	+	+	+	−	BH	GB	+	−
P54	2.8	−	−	+	+	+	−	BH	W	+	−
SW1	2.5	−	−	+	−	−	−	BH	GB	+	−
SW3	2.5	−	−	+	−	−	−	BH	GB	+	−
SW10	2.5	−	−	+	−	+	−	BH	GB	+	−

“+”—presence of the tested trait; “−“ no survey trait; BH—blue with halo; GB—green–blue; W—white; cell sizes that differ by at least 0.1 mm are statistically significant.

**Table 5 foods-12-03630-t005:** Antibiotic resistance of reference strains and atypical *L. monocytogenes* isolates (mm); name abbreviations and concentrations of antibiotics are provided in the Materials and Methods section.

Sample	Antibiotics													
	GEN 10	STR 25	KAN 30	CHL 30	RIF 5	CET 30	VAN 30	CLI 2	ERY 15	AMP 10	MEZ 75	PEN 10	CIP 5	TET 30
1/2a	25	23	30	29	32	28	30	13	33	28	31	40	25	30
1/2b	20	21	24	28	30	29	26	15	36	33	32	28	23	30
1/2c	27	23	26	27	34	28	26	21	32	39	27	31	20	27
3a	29	30	32	26	25	32	31	13	38	41	37	30	18	37
3b	26	20	26	26	28	24	26	18	28	24	24	21	22	28
3c	23	18	28	21	30	26	23	12	26	27	28	25	19	26
4a	28	22	28	24	28	26	26	12	32	31	30	28	21	30
4b	28	19	25	24	28	26	24	10	26	26	25	25	22	28
4c	22	20	28	23	26	24	25	14	29	30	31	30	19	26
4d	24	20	24	21	28	23	26	14	28	24	24	22	21	25
4e	23	25	20	24	28	20	28	15	29	40	28	24	25	26
7	25	15	27	25	26	26	28	19	31	27	25	25	20	24
112	20	17	21	28	21	21	26	12	30	21	29	29	20	27
245	23	24	23	23	22	23	21	14	30	27	23	27	22	24
273	24	20	24	29	27	26	28	22	31	32	31	29	22	28
275	23	15	24	26	27	27	27	12	26	33	31	27	23	27
280	33	28	26	31	31	20	24	0	30	25	31	23	30	12
281	23	17	21	23	25	23	23	10	29	31	28	29	22	22
288	31	22	32	29	24	16	23	10	22	19	30	22	23	12
324	23	21	23	18	20	14	22	10	26	19	17	18	27	28
352	22	20	25	26	26	24	28	13	26	30	28	27	22	22
69	21	0	23	24	25	25	23	15	28	30	26	27	17	10
111	29	24	32	25	27	18	26	0	27	20	30	24	27	15
140	22	22	25	26	26	25	22	14	32	28	28	27	17	29
B15	22	13	24	23	22	18	18	0	28	10	27	10	23	25
B54	30	27	20	31	22	26	25	25	28	31	31	27	28	20
B56	15	14	18	30	22	17	26	13	32	24	25	17	23	27
B16	30	27	20	31	22	26	25	25	28	31	31	27	28	20
RS26	24	27	13	25	23	22	21	20	25	29	27	23	26	0
RS29	26	28	10	18	13	23	24	20	25	19	27	15	29	0
B67	28	32	16	30	24	25	27	25	29	32	29	28	30	10
B68	28	30	30	26	24	32	28	18	34	39	38	32	18	34
D20	26	28	10	18	13	23	24	20	25	19	27	15	29	0
KS68	26	20	28	24	24	20	24	0	28	11	29	0	30	30
T8	27	27	21	31	32	29	26	22	24	25	26	23	21	25
T16	27	27	31	30	25	28	23	23	21	27	25	22	20	22
T18	27	26	22	32	28	24	25	21	28	36	34	33	29	27
CA11	26	25	31	29	27	29	27	21	24	28	32	23	25	14
P54	22	15	20	26	21	23	23	20	29	27	30	17	19	24
SW1	25	29	19	25	18	23	23	20	23	21	17	23	25	19
SW3	24	28	18	12	24	20	22	0	26	29	27	26	27	0
SW10	27	28	16	27	23	25	22	23	27	30	27	26	29	28

**Table 6 foods-12-03630-t006:** Occurrence of virulence genes in reference to *L. monocytogenes* and atypical *L. monocytogenes* and serological classification.

Sample	Virulence Genes														
	*hly*	*iap*	*prfA*	*inlA*	*inlB*	*inlC*	*inlJ*	*plcA*	*plcB*	*llsX*	*actA*	*mpl*	*LMOf* *2365_* *0970*	*LMOf* *2365_* *2721*	Serogroup
1/2a	+	+	+	+	+	+	+	+	+	−	+	+	+	+	nc
1/2b	+	+	+	+	+	+	+	+	+	+	+	+	+	+	nc
1/2c	+	+	+	+	+	+	+	+	+	−	+	+	+	+	nc
3a	+	+	+	+	+	+	+	+	+	−	+	+	+	+	nc
3b	+	+	+	+	+	+	+	+	+	−	+	+	+	+	nc
3c	+	+	+	+	+	+	+	+	+	−	+	+	+	+	nc
4a	+	−	+	+	+	−	+	+	+	−	+	−	+	+	nc
4b	+	+	+	+	−	+	+	+	+	−	+	+	+	+	nc
4c	+	−	+	+	−	−	+	+	+	−	+	+	+	+	nc
4d	+	+	+	+	−	+	+	+	+	−	+	+	+	+	nc
4e	+	+	+	+	−	+	+	+	+	+	+	+	+	+	nc
7	+	+	+	+	+	+	+	+	+	+	+	+	+	+	nc
112	+	−	−	−	−	−	−	−	−	−	−	−	−	+	nc
245	+	−	−	−	−	−	−	−	−	−	−	−	−	+	nc
273	+	−	+	−	−	−	−	−	−	−	−	+	+	+	1/2c−3c
275	+	−	−	−	−	−	−	−	−	−	−	−	−	+	nc
280	+	−	−	−	−	−	−	−	−	−	−	−	−	+	nc
281	+	−	−	−	−	−	−	−	−	−	−	−	−	+	nc
288	+	−	+	−	−	−	−	−	−	−	−	+	+	+	1/2c−3c
324	+	−	−	−	−	−	−	−	−	−	−	−	−	+	nc
352	+	−	+	−	−	−	−	−	−	−	−	+	+	+	4a−4c
69	+	−	+	−	−	−	−	−	−	−	−	−	−	+	nc
111	+	−	+	−	−	−	−	−	−	−	−	−	−	+	4a−4c
140	+	−	+	−	−	−	−	−	−	−	−	+	+	+	4a−4c
B15	+	−	−	−	−	−	−	−	−	−	−	−	+	+	nc
B54	+	−	−	−	−	−	−	−	−	−	−	−	+	+	nc
B56	+	−	−	−	−	−	−	−	−	−	−	−	−	+	nc
B16	+	−	−	−	−	−	−	−	−	−	−	−	−	+	nc
RS26	+	−	−	−	−	−	−	−	−	−	−	−	−	+	nc
RS29	+	−	−	−	−	−	−	−	−	−	−	−	−	+	nc
B67	+	−	−	−	−	−	−	−	−	−	−	−	−	+	nc
B68	+	−	−	−	−	−	−	−	−	−	−	−	−	+	nc
D20	+	−	−	−	−	−	−	−	−	−	−		−	+	nc
KS68	+	−	−	−	−	−	−	−	−	−	−	−	−	+	nc
T8	+	−	−	−	−	−	−	−	−	−	−	−	−	+	nc
T16	+	−	−	−	−	−	−	−	−	−	−	−	−	+	nc
T18	+	−	−	−	−	−	−	−	−	−	−	−	−	+	nc
CA11	+	−	−	−	−	−	−	−	−	−	−	−	−	+	nc
P54	+	−	−	−	−	−	−	−	−	−	−	−	−	+	nc
SW1	+	−	−	−	−	−	−	−	−	−	−	−	−	+	nc
SW3	+	−	−	−	−	−	−	−	−	−	−	−	−	+	nc
SW10	+	−	−	−	−	−	−	−	−	−	−	−	−	+	nc

“+”—presence of the target gene; “−“—absence of the gene; nc—not classified.

## Data Availability

Any data or material supporting this study’s findings can be made available by the corresponding author upon request.

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
