# Peer review of "Phenotypic and Genotypic Characteristics of Non-Hemolytic L. monocytogenes Isolated from Food and Processing Environments"

_foods, 2023, doi:10.3390/foods12193630_

Round 1
Reviewer 1 Report
The article "Phenotypic and genotypic characteristics of non haemolytic L. monocytogenes isolated from food and processing environments" presents relevant information for the generation of strategies for the identification of Listeria monocytogenes strains and their differentiation into typical and atypical strains in order to generate strategies for the prevention of listeriosis. Specific comments on the manuscript are made below:
L12 What do you mean by "The 30?" 30 isolates?
The abstract should include a sentence describing the impact of the results found.
L47 Place in italics L. monocytogenes
L83 Please indicate the year in which the soil samples were taken.
L125 The expression "based on years of research" could be eliminated from the sentence and leave only the description of the adaptation of the method, how was this adaptation arrived at? what are the results of these years of research that allow having an adequate method for the cultivation of aLM?
L136 Again, "on the basis of years of research" should be removed from the expression. how do the patent modifications respond to the needs/characteristics of non-hemolytic aLM strains for DNA isolation?
L172 How was antibotic resistance expressed?
L200-203 In these lines author discusses how there is not always a correlation between higher LM frequency and aLM frequency; however, lines 183-199 describe the origin of the 30 aLM strains found in the present work but does not show the correlation between LM and aLM isolates. Therefore, it is recommended to add this approach in the analysis of the results of the present work.
In the case of soil samples, is there any effect of season of the year on the prevalence of LM and aLM?
L207-209 Although the target of the study is aLM strains, it is important that the author discusses the incidence of LM strains and their origin, because in the discussion data on the occurrence of "typical" L. monocytogenes strains is shown. This information is also relevant.
L316-317 Was this applied on the medium adapted for aLM growth (carbohydrate profile fermentation)?
L436 Would it be possible to classify them as a new species?
L508 Place S. aureus in italics.
L516-518 This conclusion refers to information for which no results are presented or discussed in the manuscript.
In conclusion, could the author propose a strategy for more accurate identification/differentiation of aLM vs. LM strains? Is it possible to propose a subclassification of aLM strains based on their hemolysis pattern, motility, and CAMP test against S. aureus?
Except for supplementary material, Tables and Figures should be placed in the manuscrit.
Figure 1. It is suggested to add labels to the groups generated in the dendograms that allow easy identification of which types of strains are contained in each one. Alternatively, you could use different colors for the groups so that the interpretation of the dendograms is faster.
Figure 2. Remove the border of the boxes for each of the attributes analyzed.
Table 3. Place the unit of measurement of inhibition.
Table S1. In the column headings, correct "izolates".
Reviewer 2 Report
The most prominent advantage of this article is that the sample size is extremely sufficient and the types and sources of samples are very wide, which make the data representative. Moreover, the author conducted an in-depth analysis and summary of all the data. However, the manuscript still has many problems and requires further careful revision and improvement:
1. It should be highlighted in the section of Introduction and/or Abstract that the aim and significance of this study and the contribution of this study to the detection or control of L. monocytogenes in food.
2. Line 41: The location of “ [3,4] “ is some odd.
3. Line 90-92: It would be better if the author could provide more detailed information about the samples. For example, listing the names of “various retail outlets” and “13 different producers” in supplementary.
4. Line 151: It should be avoided that sentences beginning with Arabic numerals. Therefore, it had better to change “11” here to “Eleven”. It applies to all the manuscript.
5. Line 238-239: “it can be assumed that the occurrence of non-haemolytic aLMs may also be a consequence of prfA/sigB gene mutations and/or is a result of environmental stress”. It is not rigorous for the author to make such an assumption based only on the information from reference [38]. It lacks sufficient evidence.
6. All the Figures and Tables should be self-explanatory. Therefore, Tables 1, 2, and 4 should explain the meaning of symbols (“+” and “-“) in the tables. Moreover, the explanation should be placed as a footnote below the Table instead of together with the title.
7. Table 3 should provide the units and meaning of all data.
8. The author should suggest some measures for the detection and control of L. monocytogenes in different food based on the results of this study in Discussion and/or Conclusion section.
9. Please double check the format of the References.
10. According to the format requirements of Foods, authors should put all Figures and Tables (except supplementary) at the appropriate locations in the manuscript.
Minor editing of English language required
Reviewer 3 Report
This is an interesting manuscript describing a rather large investigation of food and food related materials concerning the occurrence of Listeria monocytogenes. Specifically, the author is reporting her findings concerning atypical isolates of Listeria monocytogenes. Although the subject is certainly of interest, the author did not succeed on providing the significance to public health of these atypical strains. Specifically, it should have been tested or discussed if the atypical strains can transmit resistance or more interestingly pathogenicity traits to other Listeria monocytogenes cells.
General comments.
The Results and Discussion part needs revisions.
1. It is extremely large and more suitable to a dissertation thesis. Some aspects should have been reported in summary (way of action of specific genes, morphological characteristics of strains) and more emphasis should be given to the occurrence of aLM reported by other researchers.
2. The Public Health aspects of the occurrence of aLM should have been reported in more detail.
Specific comments
P2 L50. "ferments rhamnose and does not degrade xylose”. It can been written as “ferments rhamnose but not xylose” since for both rhamnose and xylose the fermentation is examined.
P2 L61 – 65. These two sentences seem out of place, since the text previously has been specified to non – haemolytic strains.
P2 L69. “haemolytic” instead of “haemolitic”.
P2 L74. “to isolate” or “the isolation” instead of “to isolation”.
P2 L85-86. The soil samples were not described in detail. Specifically, the samples sites and the criteria for selecting the sites were not reported.
P3 L109 – 111. No justification nor a reference to this procedure has been given.
P3 L117 – 119. A reference is needed.
P4 L164 – 172. The criteria for inclusion of these antibiotics and for the interpretation of the inhibition zones have not been reported.
P4 L184. This table is essential for the understanding of the manuscript. Please add it in the main text.
P5 L 204-215. Please summarize this paragraph in a couple of lines and merge with the previous paragraph.
P5 L217. Although motility is reported, it has not been described In the Materials and Methods part.
P5 L 240. As with motility, the CAMP test has not been described. In addition, it is rather peculiar how did the strains show a positive result against S. aureus in CAMP test (enhancement of S. aureus haemolysis) without showing any haemolysis. Can the author provide a photo of this?
P6 L246 – 252. This is a repetition of the Materials and Methods. It should be reported in more detail in this part, not here.
P6 L252-254. The correlation of the findings in non pathogenic Listeria (L. innocua) to possibly pathogenic Listeria monocytogenes is risky.
P6 L 256 – 262. Differences in size should be checked statistically to verify if there is a difference.
P6 L263-278. The reported results and the hypotheses concerning the aLM size can be the subject of a new research project, involving methods screening the bacterial metabolism of these strains. This part should be reduced to a minimum (3 – 4 sentences), keeping the possible explanations at a minimum.
P6 L 280. No information on the dendrogram construction methodology.
P7 L329. Statistical grouping has not been reported in the Materials and Methods part.
P7 L341-343 and elsewhere. The exact size of the zone of inhibition is not of interest. Several factors can affect, and small differences are usually present. The strains can not be categorized as more or less resistant according to the size of the inhibition zones.
P8 L 360 – 362. A rather common phrase not adding to the manuscript. Please omit.
P8 L369. Please omit “among”.
P8 L377. A-C of Fig 1C is not in the main manuscript, but in the supplementary files.
P8 L387-8. “…in strains 4a and 4c”. This should be substituted by the type strains names used, both in the text and the manuscript.
Several parts of the manuscript requested by the publisher are missing (e.g. Acknowledgements, Contribution, etc.).
Supplementary material. At least half of these tables and figures deserve a place in the manuscript, preferably Figures 1 and 3, and Tables S1, 1, and 4. The results in Table 3 are of interest but not in this form. The report of the resistance to specific antibiotics should be reported.
Figure 1. The cut-off value used is variable between the different dendrograms. Can you comment on this?
Figure 2. Difficult and very understandable PCA. Please keep a supplementary file.
Table S1. Please remove the zero values in order to be clearer.
Table S2. In some cases, there is no reference (e.g. inlB and inlC). Please revise or report the methodology for primer design.
Table 1. Since they are the same in all strains, both type strains and aLM, the columns reporting Esculin test, Voges-Proskauer test, Methyl Red test, Nitrate reduction test, and growth at 30oC and 37oC should be omitted.
Round 2
Reviewer 2 Report
The manuscript was revised well accordingly.